# Barriers to the hospital treatment among Bede snake charmers in Bangladesh with special reference to venomous snakebite

Ken Yoshimura[1,2]*, Moazzem Hossain[3]*, Bumpei Tojo[4], Paul Tieu[5,6,7], Nathalie Nguyen Trinh[6], Nguyen Tien Huy[1,7], Miho Sato[1,8], Kazuhiko Moji[1,8]*

**1** School of Tropical Medicine and Global Health, Nagasaki University, Nagasaki, Japan, **2** Japan Snake Institute, Gunma, Japan, **3** Institute of Allergy and Clinical Immunology of Bangladesh, Dhaka, Bangladesh, **4** World Language and Society Education Centre, Tokyo University of Foreign Studies, Tokyo, Japan, **5** McMaster University, Ontario, Canada, **6** University of Toronto, Ontario, Canada, **7** Online Research Club (https://www.onlineresearchclub.org/), Nagasaki, Japan, **8** School of Global Humanities and Social Sciences, Nagasaki University, Nagasaki, Japan

\* yoshimura@snake-center.com (KY); moazzem.iacib@gmail.com (MH); moji-k@nagasaki-u.ac.jp (KM)

**Data Availability Statement:** All relevant data are within the manuscript and the supporting information.

## Abstract

Snakebite envenoming is a potentially life-threatening global public health issue with Bangladesh having one of the highest rates of snakebite cases. The Bede, a nomadic ethnic group in Bangladesh, traditionally engages in snake-related business such as snake charming. The Bede relies on their own ethnomedicinal practitioners for snakebite treatment while there is a lack of concrete evidence on the effectiveness of such ethnomedicinal treatment. To identify the barriers to the utilization of biomedical treatment for snakebite we conducted interviews with 38 Bede snake charmers, who have experienced snakebite, and six family members of those who died of snakebite. Our results show that four critical barriers, Accessibility, Affordability, Availability, and Acceptability (4As), prevented some of the Bede from seeking biomedical treatment. Moreover, we found that a few Bede died of a snakebite every year. There are survivors of snakebite who were able to receive biomedical treatment by overcoming all of the 4As. Our results provide insights into the current state of snakebite treatment in Bangladesh and can inform the development of more effective and accessible treatment options for those affected. Partnership between the public sector and the Bede community has the potential to make a significant impact in reducing snakebite morbidity and mortality in Bangladesh.

## Author summary

Snakebite envenoming is a life-threatening issue. The Bede people in Bangladesh often engage in the snake business with professions such as snake charmers and snakebite healers, however, this makes them more vulnerable to snakebites. There is little research on health seeking behavior and its outcomes of the Bede people. Bangladesh has the highest rate of snakebites in the world where snakebite victims (both the Bede and non-Bede people) typically rely on the ethnomedicinal treatment provided by the Bede healers. Hence,

**Funding:** The author(s) received no specific funding for this work.

**Competing interests:** The authors have declared that no competing interests exist.

clarification of their health seeking behaviors will contribute to the overall improvement of snakebite treatments among the Bede people and the rest in Bangladesh. In depth interviews were conducted, and conventional content analyses were performed. Although 33 of the 38 snake charmers answered that hospital treatment is the best way to care for snakebite, only five went to the hospital for treatment. Among 44 snake charmers, only one successful antivenom treatment was reported in this study. The accessibility to antivenom treatment is limited for the Bede community despite their proximity to the snake. To make antivenom treatment available within the Bede community, collaboration between the public sector and the Bede ethnomedical practitioners should be promoted.

## Introduction

Envenoming by snakebite is often a life threatening disease caused by the toxins of a venomous snake [1]. Of the estimated 5.4 million people bitten by snakes each year globally, 2.7 million are thought to be envenoming. As a result, 81,000–138,000 people die from complications caused by envenoming. Even if they survive, many are left with amputations and other permanent disabilities [1]. However, the matter has been disregarded in the global health agenda until the World Health Organization recognized snakebite envenoming as a Neglected Tropical Disease (NTD) in 2017. South Asia has the highest rate of snakebite envenoming in the world, which contributes to 70% of global snakebite mortality [2].

While some snake species are influenced by climate change [3,4] the number of snakebite increase during monsoon season as flood, cyclone, storm surge, and bank erosion drive snakes out of their usual habitats and force them to migrate to near the residential areas of humans [5, 6]. The number of snakebite cases in Bangladesh is estimated at 623/100,000 persons per year, which is one of the highest rates in the world [5]. Out of the 82 species of snakes that populate this region, 27 are venomous and 6 are venomous which are not docile and likely to strike actively [6].

There are several factors causing the high rate, for example, a high population density, extensive agricultural activities, diverse venomous snake species, and lack of efficient snakebite control programs [7]. The venoms used by snakes are a part of their self-defense mechanism in response to a threat. Snakebite by the Viperidae (e.g., pit vipers) and Elapidae (e.g., kraits and cobras) are especially harmful to people [2]. Because of the wide scope of symptoms ranging from local tissue damage to acute renal failure, treatment of venomous snakebite requires appropriate medical facilities [7]. In contrast to most NTDs, the medical treatment for snakebite envenomation is time-critical, requiring an appropriate selection of antivenom by adequately trained staff in a well-equipped hospital [8]. However, many victims do not seek biomedical care but from traditional snakebite healers. Generally, and globally, this situation can be explained as "patterns of resort," which explains typical health seeking behavior; that the first health seeking option tends to be the most basic and cheapest. When it is unsuccessful, people seek more advanced, costly, and unconventional treatment, notably in Lao PDR [9], India [10], Nigeria [11], and South Africa [12] In terms of snakebite treatment, in Bangladesh, Rehman et al in 2009 reported that 86% of the snakebite victims' first contact was with snakebite healers or snake charmers (snake handlers) after snakebite [5]. A different 2003 study by Hossain et al observed that approximately 61% sought treatment from traditional snakebite healer or herbal medicine practitioner [13]. However, it is generally understood that ethnomedicinal treatment by traditional healer's lack of medical evidence.

Traditional healers known as *Ojha* (*Ozha*), who are mostly Bede and have been known to use potentially harmful methods such as the improper use of tourniquet and multiple incision around the bite site for oral blood suction. In a significant proportion of cases, these methods have been used inappropriately and are not scientifically approved [5,14].

Hence the outcome of traditional approaches of using a tourniquet, creating incisions at the bite site, applying herbal products, and various other rituals are determined by chance [15]. The *Ojha*, which remove snake venom from a snake-bitten person by means of a traditional method, is important in the Bede community [16].

Based on an analysis of the available literature, Brandt reports the Bede as follows: (1) the Bede's total population in Bangladesh is estimated to be 1.2 million, (2) Bede is a nomadic community, (3) the Bede people are a distinct social and cultural group, (4) they can be referred to as an ethnic group or "tribe", (5) they live on boats on rivers (hence also known as "river gypsies"), (6) they are poor, socially excluded, and have no basic rights, (7) their culture is under threat due to forced sedentarization and (8) they often earn their income through snake related occupations, such as snake charming, snake catch, and healing snakebite (*Ojha*) [17].

Snakebite is a critical issue for the Bede people, however, there is little scholarship on the health seeking behavior of the Bede people regarding snakebite. As research on a Bede community conducted by Hossain et al revealed, 59.6% (134/225) of the respondents sought the help of biomedical treatment for snakebite, while the following 19.6% (44/225) used the traditional Bede treatment [16]. However, the research did not take the itinerant/nomadic lifestyle of the Bede into account. The options for treatment vary according to the location of snakebite incidents (e.g., whether the Bede was settled or nomadic, whether the snakebite occurred in a rural area or in an urban area). Therefore, the settlement patterns of the Bede study participants must also be considered. Savar, our study site, is located on the outskirts of Dhaka, which has relatively easy access to hospitals, especially if the Bede quit their nomadic lifestyle and settle down.

Due to their snake related professions, many Bede people are prone to snakebite. Thus, we aim to clarify the health seeking behavior of the Bede people regarding snakebite and its outcome. The research findings are expected to help to overcome barriers to hospital treatment and provide insights into the health seeking behavior of the Bede people regarding snakebite.

## Methods

### Ethics statement

Ethical approval was obtained from the School of Tropical Medicine and Global Health, Nagasaki University (approval number is NU_TMGH_2020-091-0) and the National Research Ethics Committee of the Bangladesh Medical Research Council (Registration Number is 267 24 11 2019). The Bede are ethnic minorities and vulnerable in society; thus, the PI first explained to their leaders the purpose and procedure of our study and their consent was obtained. The Institute of Allergy and Clinical Immunology of Bangladesh (IACIB), which has a hospital in Savar, supported the implementation of this research. The organization has a history of providing free health services to the Bede community, including snakebite treatment, from 2014 to 2019, with support from various organizations.

Written informed consent from all participants were obtained (S1 File).

In addition, the process of data collection and the right to decline participation were explained. A printed information sheet, including contact information of the researchers and the IACIB, had also been provided before we obtained the written informed consent. All printed data collected were stored in a locked cabinet in a secured room in the IACIB. All digital data were saved in the PI's laptop with a password, as well as the PI's laptop was stored in a locked cabinet in a secured room.

## 1. Study setting

The study was conducted in Pura village, Boktapur (Bakterpur) Union, and Savar Upazila of Dhaka District in Bangladesh (Fig 1). The population was about 12,000 [18]. There are approximately seven thousand five hundred Bedes in Pura village [16], the biggest Bede community in the country. Despite some of them now having their own houses along the banks of the Bangshi River, they still continue their way of life by periodically traveling via boat to different villages, away from their homes, and engaging in various traditional occupational pursuits [17,18].

## 2. Study design and study period

The study took a qualitative approach (S2 File). In-depth, semi structured interviews and key informant semi structed interviews were conducted by Principal Investigator (PI), fluent in Bengali, and two Research Assistants (RAs), who are native speakers of Bengali, using an interview guide (S3 and S4 File).

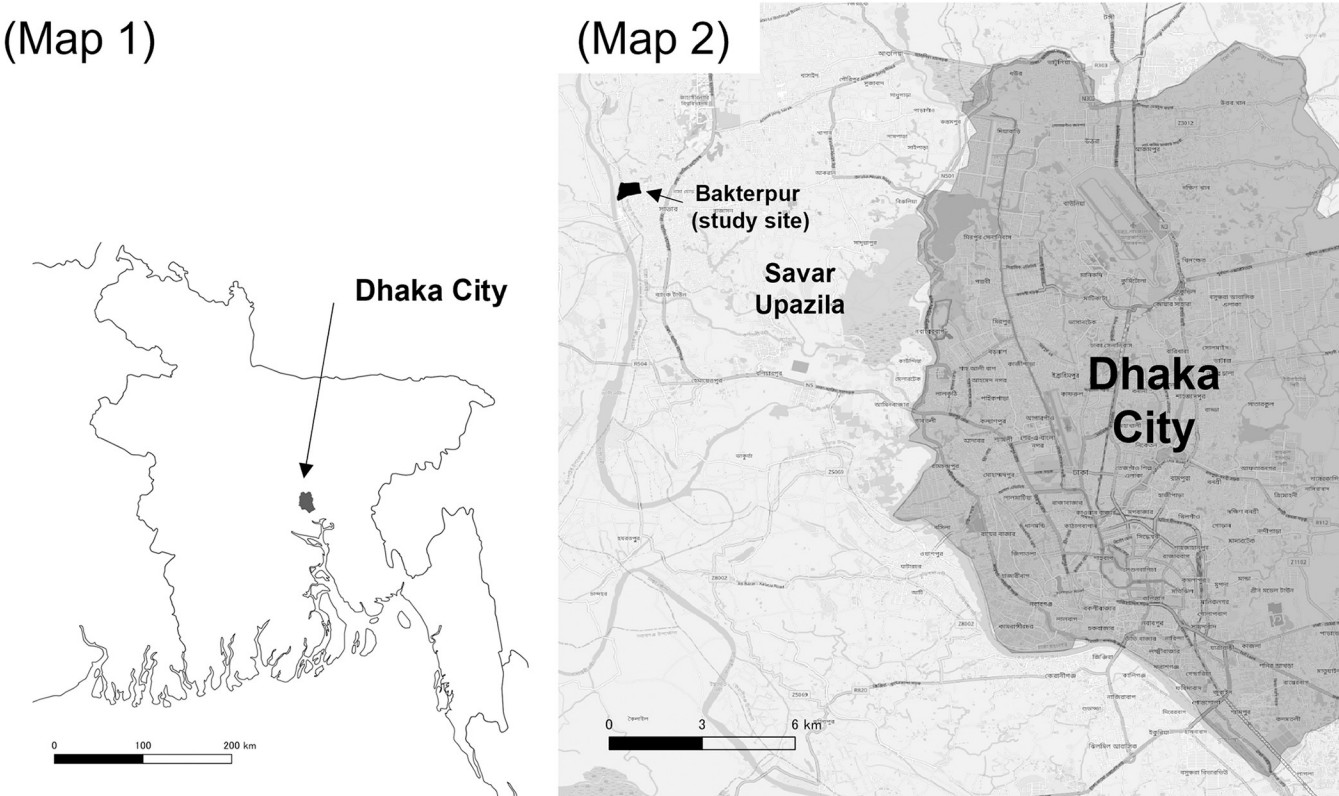

**(Map 1) Bangladesh showing Dhaka City.          (Map 2) Dhaka to Bakterpur.**

Map was produced using QGIS Version 3.22.14 (QGIS Development Team, 2021).
Source of shapefile (Map 1):
          https://www.naturalearthdata.com/downloads/10m-cultural-vectors/
Source of Background map (Map 2):
          https://www.openstreetmap.org/ (https://www.openstreetmap.org/copyright/en)

**Fig 1. Map of the study site.** (1) Map of Bangladesh showing Dhaka Zila. (2) Dhaka to study site (Bakterpur).

The two assistants: one works at the IACIB and the other is a Bede person who used to work as a snake charmer from the Bede village. The two research assistants received comprehensive training to assist in this study, including briefing on sharing posters of the research and interview guide for this study, the purpose of the research, training on interview and qualitative data analysis skills, Bede's background, and ethical considerations.

Data were collected over March 2020. Considering the minority status of the Bede, prior to the study period, PI and RA from IACIB visited the Bede community and built a good relationship with the Bede [19], (S5 File).

Interviews were conducted in Bengali, were recorded only if the participant agreed, and were conducted either at a private office or at the participant's house.

Mainly PI performed those interviews and RAs supported the interviews to understand the local language and terminology for snakebite healing. Codes were extracted from each interview and were analyzed for sub-categories/categories that affect the health seeking behavior for snakebite (S6 File).

With the support of Bede leaders, participants were introduced to the snowballing sampling, which was subject to bias due to the difficulty of controlling sample composition, but a standard method for qualitative research on difficult to reach populations [20]. Those who had experienced snakebite in the last three years were invited to participate in the study.

The interviews were conducted with 38 adult Bede snake charmers who survived snakebite as well as one relative of each of the six Bede snake charmers who died from venomous snakebites.

This approach allowed us to gather insights from both survivors and family members affected by snakebite within the Bede community. Twenty Bede snakebite healers (*Ojha*) were also invited into the study to clarify their traditional therapeutic approaches to snakebite.

Snakebite marks were also checked if the marks remain to identify the snake species [21]. The interview with new participants continued until data reached saturation [22]. The interview guide was pretested to ensure its contents were appropriate during our preliminary visits to the Bede community.

## Data analysis

The verbal transcripts were translated into English. PI and two RAs reviewed every transcript for content and completeness, then coded followed by the conventional content analysis, in which codes are sorted into emergent sub categories then the cluster of sub categories form a number of categories [23]. PI and RAs confirmed multiple times the validity of the categories that emerged from the analysis with different Bede groups at the different duration of snake bite experience for triangulation [23,24].

In our manuscript, we have discussed various treatment methods used among the Bede people for snakebite, including herbal medicine, incision, suction, hospital treatment, and antivenom. We acknowledge that these treatment methods represent the pluralistic medical system, a mix of traditional, folk, and biomedical approaches among the Bede community. Herbal medicine, incision, and suction are commonly considered as traditional or folk remedies. These practices often reflect traditional healing practices within the community.

On the other hand, treatment at hospitals, which involves medical care provided in a hospital setting with the use of antivenom, are categorized as biomedical approaches. These treatments rely on clinical evidence, which comes with medical advancements to manage snakebite cases effectively.

## Results

The characteristics of study participants are presented in Table 1. The average age of the survival cases was 41.5 (range: 18–68). The average age of the six death cases was 44.1 (range: 35–65). Of the total 44 cases, 42 were identified as Muslims. Thirty-six were born in Pura village, where this study was conducted. Twenty-six of the 38 snake charmers who have been bitten by snake had a second job.

## Venomous snakebite experience of snake charmers (38 survival cases)

Presented on Table 2, among 38 survival cases, 31 were bitten during their snake charming performance, six were bitten during catching snakes, and one was bitten during feeding snakes. Although Elapidae was the perpetrator of most of the attacked except for four cases by

**Table 1. Background/characteristics of snake charmers (survival cases, death cases) and traditional healers.**

|  | Survival cases (N = 38) | | Death cases (N = 6) | | Traditional Healers (N = 20) | |
|---|---|---|---|---|---|---|
|  | Frequency (N) | Percent (%) | Frequency (N) | Percent (%) | Frequency (N) | Percent (%) |
| **Sex** | | | | | | |
| Male | 33 | 87 | 6 | 100 | 11 | 55 |
| Female | 5 | 13 | | | 9 | 45 |
| **Age** | | | | | | |
| 18–30 years | 10 | 26 | | | 7 | 35 |
| 30–39 years | 10 | 26 | 2 | 33 | 9 | 45 |
| 40–49 years | 4 | 10 | 3 | 50 | | |
| 50–59 years | 6 | 16 | | | 4 | 20 |
| Above 60 years | 8 | 21 | 1 | 16 | | |
| **Religion** | | | | | | |
| Muslim | 37 | 98 | 5 | 83 | 19 | 95 |
| Hindu, Other | 1 | 2 | 1 | 16 | 1 | 5 |
| **Educational status** | | | | | | |
| No education | 21 | 55 | 5 | 83 | 15 | 75 |
| Some /complete primary | 16 | 42 | 1 | 16 | 5 | 25 |
| Complete secondary | 1 | 3 | | | | |
| **Hometown** | | | | | | |
| Pura village | 31 | 81 | 5 | 83 | 11 | 55 |
| Other | 7 | 19 | 1 | 16 | 9 | 45 |
| **Nomad or settled** | | | | | | |
| Nomad, Semi-nomad | 19 | 50 | 6 | 100 | 11 | 55 |
| Settled down | 19 | 50 | | | 9 | 45 |
| **Moving route** | | | | | | |
| Bangladesh | 13 | 34 | 3 | 50 | | |
| India, other | 25 | 66 | 3 | 50 | | |
| **Monthly income (Tk)** | | | | | | |
| 0–5,000 | 7 | 18 | | | | |
| 5,001–10,000 | 25 | 65 | | | | |
| Over 10,000 | 6 | 17 | | | | |
| **Second job** | | | | | | |
| Yes | 26 | 68 | | | | |
| No | 12 | 31 | | | | |

**Table 2. Venomous snakebite experience of snake charmers (survival cases and death cases).**

| | Survival cases (N = 38) | | Death cases (N = 6) | |
|---|---|---|---|---|
| | Frequency (n) | Percent (%) | Frequency (n) | Percent (%) |
| **Where** | | | | |
| Roadside | 26 | 68 | 1 | 17 |
| Field | 10 | 26 | 4 | 66 |
| Room | 2 | 5 | 1 | 17 |
| **What were you doing then?** | | | | |
| Handling snakes (snake charming performance) | 31 | 81 | 5 | 83 |
| Catching snakes | 6 | 15 | | |
| Feeding snakes | 1 | 2 | 1 | 17 |
| **Bite site** | | | | |
| Hand | 27 | 71 | 2 | 33 |
| Arm | 6 | 15 | 2 | 33 |
| Leg | 2 | 5 | 2 | 33 |
| Foot | 2 | 5 | | |
| Chest | 1 | 3 | | |
| **Type of snake** | | | | |
| Indian cobra | 18 | 47 | 2 | 34 |
| King cobra | 13 | 34 | 4 | 66 |
| Green pit viper | 4 | 10 | | |
| Monocled cobra | 3 | 7 | | |
| **What do you think is the best treatment for snakebite?** | | | | |
| Treatment at a hospital | 30 | 78 | | |
| Traditional treatment | 8 | 21 | 5 | 83 |
| No treatment | | | 1 | 17 |
| **Symptom※** | | | | |
| Neurotoxic sign | 7 | 18 | 5 | 83 |
| Bleeding | 31 | 81 | | |
| NA | 4 | 10 | 1 | 17 |
| **Snakebite coping** | | | | |
| Treatment at a hospital | 5 | 13 | | |
| Traditional medicine (Ojha) | 30 | 78 | | |
| No treatment | 3 | 8 | | |
| **Barrier for the treatment at hospital?** | | | | |
| Distance | 26 | 68 | | |
| Cost | 12 | 31 | | |
| **Prevention for snakebite** | | | | |
| Defang | 22 | 57 | | |
| Removal of venom gland | 3 | 7 | | |

※ Some have multiple symptoms

green pit viper (*Trimeresurus albolabris*), the symptoms varied, not only the neurotoxic symptoms, but also bleeding symptoms. The identified Elapid are Indian cobra (*Naja naja*), King cobra (*Ophiophagus hannah*) and Monocled cobra (*Naja Oviparous*) (Fig 2).

Despite 30 of the 38 answered that hospital treatment is the best way to care for snakebite, only five actually went to the hospital for the treatment.

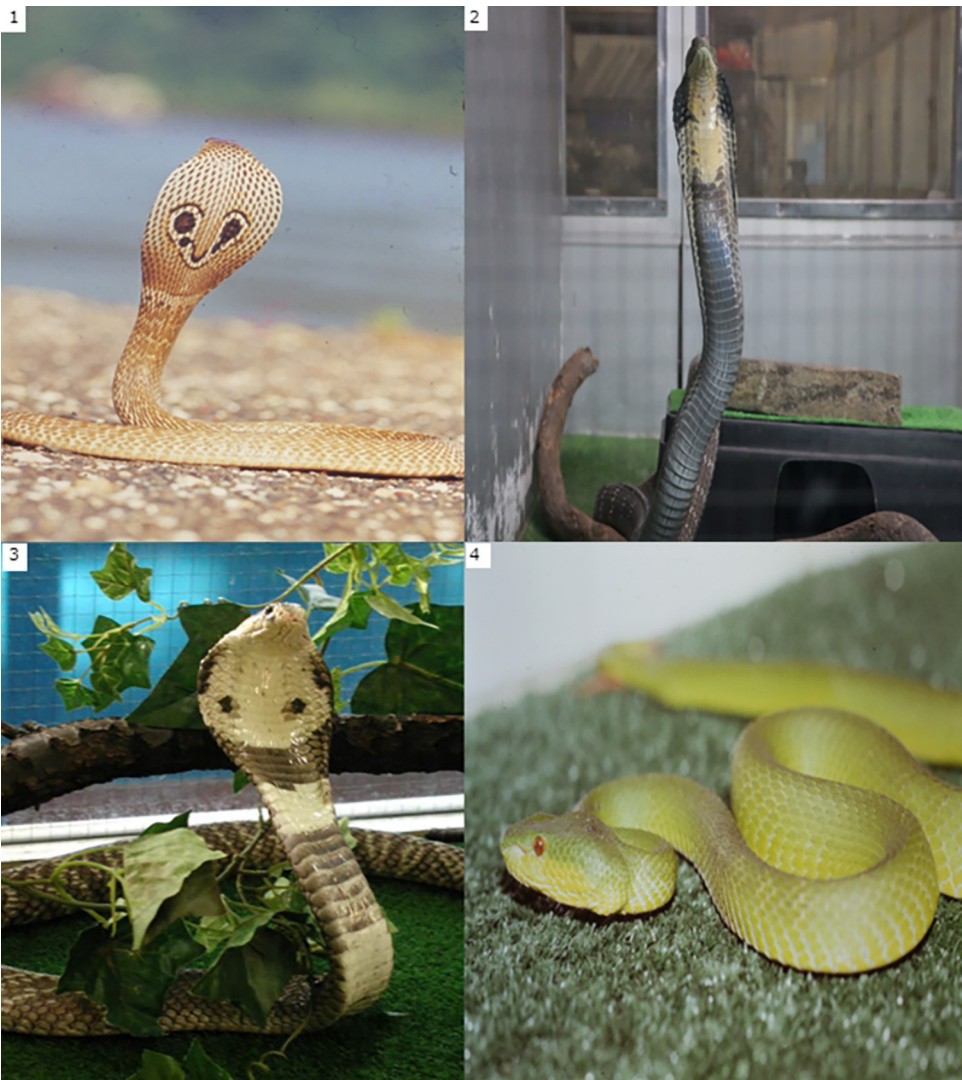

**Fig 2. Bitten by Four Venomous Species.** (1) Indian cobra (*Naja naja*) (2) King cobra (*Ophiophagus hannah*). (3) Monocled cobra (*Naja Oviparous*) (4) Green pit viper (*Trimeresurus albolabris*).

Notably, one victim recovered dramatically after the hospital treatment. This male snake charmer was bitten in the Pura village. He was immediately taken to a hospital in Dhaka by motorcycle. He was treated with antivenom and hospitalized for a week at the Intensive Care Unit.

Twenty-six of the 38 answered that distance is the most challenging barrier to modern medicine. The cost of treatment was the second most challenging barrier, answered by 12 participants. Thirty participants were treated by traditional Bede treatment, 21 with tourniquets, 14 with herbal medicine, and 3 with both combined (Fig 3).

For the prevention of severe consequences from snakebite, 22 out of 38 removed the fangs from their snakes and three removed the venom glands. However, even if the fangs were removed, the participants reported snakebite experience. One explanation is that they were bitten by growing fangs, or they were bitten when catching snakes or removing their fangs.

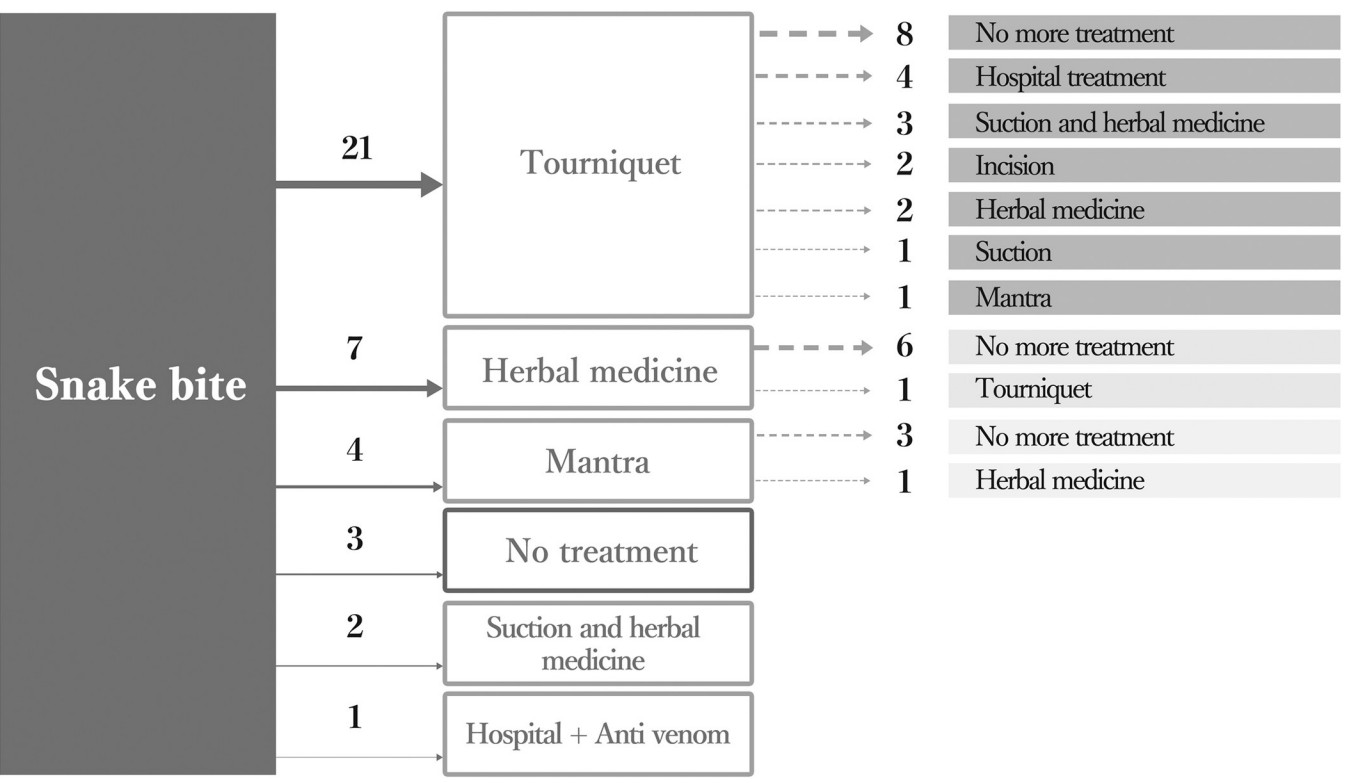

**Fig 3. Treatment choices of 38 snakebite victims (survival cases).**

## Venomous snakebite experience of snake charmers (six death cases)

As presented on Table 2, the location of snakebite varied; one case was in Pura village, one in India and four cases in rural areas of Bangladesh (Rangpur, Rajshahi, and Barisal).

Five cases occurred during a snake charming performance (one on the roadside and four in the field), while the remainder occurred during feeding in the room. All snakes belonged to the Elapidae; two Indian cobras and four king cobras. While five victims received traditional snakebite treatment from traditional healer, one victim sought no treatment. None was able to go to a hospital for treatment. All victims showed neurotoxic symptoms such as numbness, muscle paralysis, and dyspnea. Five victims' condition rapidly worsened, and they died within a few hours after the snakebite. In the remaining cases, the victim was alone at the time of death and no accurate time was confirmed.

## Key informant interview

A total of 290 codes were generated from the verbal transcripts. These codes were then organized into emergent sub-categories, ultimately forming seven main categories: "Treatment option," "Factor effect treatment," "Occupational risk/disease," "Preventive measures," "Impact on life," "Animal welfare," and "Cause of death." Each category contributed valuable insights into various aspects of snakebite and its impact on the Bede community.

Among these categories, our focus was primarily on "Factor effect treatment," which directly addresses our objective of examining the health seeking behavior of the Bede people concerning snakebite and the outcomes. However, we also recognized the significance of other

categories, such as "Occupational risk/disease" and "Impact on life," in relation to our research objective.

The Bede snake charmers in Bangladesh operate within a unique socio-economic context that presents specific challenges and limitations. Primarily, they face limited alternative job options, making it difficult for individuals to transition away from their snake-related professions.

> *The Shapuria* [snake charmer] *is a professional job and passed down from generation to generation, so we cannot change this job. We are non educated people, we cannot get other jobs. There is no option to do other work. (Snake Charmer, 30s, male; Snake Charmer, 20s male)*

Over time, there has been a noticeable decline in snake related professions and practices, further exacerbating the challenges faced by the Bede snake charmers in finding alternative means of livelihood.

The decline in snake related professions limits their job prospects and limits their job prospects and impacts their financial stability. With a reduced demand for snake charming and related services, they struggle to generate sufficient income, leaving them with little choice but working within this occupation.

> *What else can I do? I am poor and helpless. I cannot do any other job, and I need to provide for my family. I am not educated enough to pursue other occupations. (Snakebite healer, 30s, female)*

> *Yes, I continue to work as a snake charmer. There is no choice, cannot help it. (Snake Charmer, 20s, male)*

Due to limited alternative job options and a decline in snake related professions and practices, snake charmers find themselves compelled to continue working in this field despite the inherent risks involved.

However, the snake charmers are prone to snakebite due to the nature of their occupation. Contact with snakes is an unavoidable aspect of their work, leading to a high incidence rate of snakebite.

> *We play with snakes, catch snakes, sell snakes and dance with the snakes. We are Shapuria. If you play snakes every day, it is usual that they bite you. (Snake Charmer, 40s male)*

> *Snakebite incidence is high in this place (Pura village) because we go to many places and return here with snakes we caught. So sometimes snakes bite us, when we are playing or feeding. (Snake Charmer, 30s, male)*

To cope with the snakebite incident, they rely on traditional herbal medicine for treatment.

> *As I was bitten by a snake, I ate tree medicine and became well. The tree is good for me, as the poison has gone. That's all. (Snake Charmer, 20s, female)*

> *After that* [snakebite], *I was upset as it was the first time I was bitten by a snake. I returned to my home and told my parents that I had been bitten by a snake. They said nothing would happen as herbal medicine were working in my body. As I took it regularly, I felt well. (Snake Charmer, 30s, male)*

However, there is an availability barrier to accessing herbal medicine, making it challenging for them to obtain it when needed due to urban development and deforestation. According to them, it is available only in hill track areas in Bangladesh or some places in India.

*Medications to snakebite is different, they depend on types of snakes and venom symptoms. Mostly these medications are herbal pastes, such as flame lily [Gloriosa superba), Dutchman's pipe [Aristolochia], Indian snakeroot [Rauvolfia serpentina). But many of the traditional herbs are difficult to find due to the rural development and cash crop agriculture. (Snakebite healer, 30s, female)*

*We cannot go to the places where we could get herbal medicine for snakebite. I do not get my own medicine* [for snakebite] *now because I cannot go to India. It costs a lot.*

*The herbal places are beyond Assam, Laxmipur, Guwahati, and Andaman Island. So, if I don't have them, I go to the hospital. (Snakebite healer, 60s, female)*

One *Snakebite healer* mentioned that he imports the herbs from South Africa.

*I import the herbs* [Rauvolfia serpentina] *from South Africa in cooperation with my friend who works in the army as it is difficult to obtain the herbs here. We cannot work without them. (Snakebite healer, healer 30s, male)*

As a result, some of them resort to other traditional treatment methods, such as suction and incision.

*We mostly use herbs, a tourniquet to prevent venom circulation, and a mantra* [reciting spells]. *But now we cannot use the herbs because it is not available. Therefore, we suggest the victims go to the hospital after applying a tourniquet as the first treatment. (Snakebhte healer, 60s, female)*

Although most answered that hospital treatment is the best way to care for snakebite, there are barriers to hospital treatment for snakebite: (geographic) Accessibility, Affordability, Availability, and Acceptability.

## 1) (geographic) Accessibility

Many mentioned that distance to the hospital is the most challenging barrier to modern medicine. This is also mentioned in Table 2. Their nomadic lifestyle also influenced this barrier.

*We would bring him to the hospital, but the hospital was in Dhaka which is over 3 hours. (Snake Charmer in a death case, 30s male)*

*I was born and grew up here* [Pura village]. *At the age of six, I started to go around the country. I come back here every Eid. I know this place well. But I don't know where hospitals exist in areas I travel. (Snake Charmer, 30s, male)*

Some of them expressed that the lack of transportation to the hospital is another problem.

*We don't have a car or motorcycle, so we cannot go to the hospital far away. This is especially worse when we visit rural areas. And due to traffic jams, we cannot go to the hospital easily. (Snake Charmer, 20s, male)*

One snake charmer mentioned that there is significant difference between the Bede who settled down and the nomadic Bede.

*We are Bede who have settled are getting far better. We have access to better health facilities and our children have educational opportunities, compared to the other nomadic Bede. They cannot access immediate medical facilities, in case of health problems, child diseases and sudden complications. (Snake Charmer, 40s, male)*

## 2) Affordability

Most of the snake charmers could not afford to pay for medical care at a hospital.

*We have no money to go to a hospital. They require a lot* [for treatment]. *We will have to gather money from my family, relatives and others in case of snakebite. (Snakebite healer, 20s, female)*

*It costs a lot for snakebite treatment. Because we are poor, we cannot pay it. Also, our snake demands* (work as snake charmers and traditional healers) *have decreased compared to the past. It needs 20000 Tk or more* [for treatment]. *It is difficult or impossible* [to pay treatment fees]. *(Snake Charmer, 60s, female)*

Also, most of them did not have their health insurance.

*No way. Most of us don't have health insurance, so the treatment costs more. (Snake Charmer, 20s, male)*

## 3) Availability

Adequate treatment could not be given at a hospital due to the lack of antivenom drugs and medical equipment.

*Our first choice is hospital treatment. So, I go to the hospital immediately if I am bitten by a snake. The antivenom injection is now available in the hospital. (Snake Charmer, 60s, male)*

*The best way* [to treat snakebite victims] *is to go to the hospital or medical facilities. We can get appropriate medicine. (Snake Charmer, 20s, male)*

*I want to send a victim to the hospital if they show critical symptoms. But you know antivenom is actually hardly found in hospitals in rural areas. (Snakebite healer, 20s, male)*

One snakebite healer mentioned that it is difficult to find a hospital where victims can receive treatment for snakebite.

*We rarely receive an antivenom injection at hospitals in Bangladesh and India. Most hospitals do not supply the antivenom injection. (Snakebite healer, 20s female)*

The other two snake charmers mentioned that the opening hours of hospitals are also a problem.

*There was a hospital in the area where I was bitten by a snake, but it was closed at night. There are small community clinics in rural areas, but the problem is the opening hours, it is closed at night. (Snake Charmer, 30s, male)*

*I went to a community clinic in the evening, but it was closed at the time. (Snake Charmer, 20s, male)*

## 4) Acceptability

Many Bede people felt stigmatized and discriminated against when they received medical care at modern medical facilities. This feeling deterred them from hospitals.

*We face stigma sometimes in a hospital, pharmacy, or wherever so I don't want to go there. Many people hate and neglect us, we are looked down. They think the Bede are the low caste people. (Snake charmer, 20s, male)*

*They do not understand us, Bede. Sometimes we are told that Bede eat snakes, have a dirty job, smell bad, and are homeless. (Snake Charmer, 20s, male)*

## Diagnosis (Traditional venom test)

In addition, their health seeking behavior is intricately connected to their diagnostic practices when it comes to snakebite. Rather than relying solely on medical expertise, Bede snake charmers and traditional healer employ their traditional venom test, also known as the chili test or chicken test, to determine the venomous nature of a snakebite.

Upon a snakebite, Bede snake charmers initially assess the victim's face and observe symptoms to make an initial judgment regarding the venomous nature of the bite. Subsequently, they examine the fang mark and conduct the traditional venom test. This diagnostic approach is deeply rooted in their cultural and traditional beliefs.

*With a lot of experience, we usually understand whether the venom is venomous or not only by looking at the face and the symptoms. (Snakebite healer, 20s, female)*

*There must be a sign of two fangs at a certain distance if the victim was bitten by a cobra. (Snakebite healer, 20s, male)*

*A cobra bite must be recognized whether the bite is venomous or not. Then we use a blade to cleave the bitten area and put chili powder or salt there. You know, if the snakebite is venomous, these elements do not create pain. Then, we can understand. (Snakebite healer, 40s, male)*

*The victim is recommended to eat chili, if he doesn't feel any taste, the bite is venomous. (Snakebite healer, 20s, male)*

One of them mentioned that if there is a chicken, they use it for a test.

*Catch a chicken, place the anus of the chicken on the bite site, and keep it that way for some time. If the chicken dies, the snake is venomous. If the anus of the chicken is placed attached to the bitten area, the venom also attacks the chicken, and the chicken dies. Thus, I see whether the snakebite is venomous. (Snakebite healer, 20s, male)*

## Dry bite

In the context of snakebite among Bede snake charmers, it is important to distinguish between venomous bite and dry bite where no venom enters the victim's body. This distinction

significantly influences their health-seeking behavior. When a snakebite is determined to be a" dry bite," with no venom being injected, Bede snake charmers perceive it as a less severe situation. In such cases, their health-seeking behavior may be minimal or absent, as they do not perceive the need for medical intervention or traditional remedies.

*I've been bitten by cobras and other snakes a lot of times, but I have never been poisoned because the snakes had no fangs. Yes, I have no symptom every time. (Snake Charmer, 20s, male)*

Practically, in consequence of the dry bite, the average number of times they were bitten by a snake was high. Nevertheless, some did not have the symptoms of poison.

*I was bitten by a snake over 70 times. but I don't have any problems. (snake charmer 60s male)*

The presence of barriers to hospital treatment and the unavailability of herbal medicine have had significant consequences for the Bede community, resulting in unfortunate outcomes such as fatalities due to snakebite envenomation.

*A lot of people have died before, but even now two, three people among us have died because of snakebite every year. (Snakebite healer, 20s, female)*

Ten snake charmers mentioned that their relatives or family members have died because of snakebite.

*My uncle, grandfather, other relatives died from snakebite. Just six months ago, one neighbor, a young boy, died. (Snake charmer male 30s)*

### After effects

Furthermore, it is important to highlight that snakebite incidents among the Bede snake charmers lead to fatalities and various after effects for those who survive.

*My life has changed due to the snakebite. I was bitten here* (hand). *Because of the bite, I still feel pain, and some of my fingers have not been straight. (Snake Charmer, 20s, male)*

*I had a headache and a little dizziness after I was bitten by a snake. I did not feel good, I have lost all interest in it, I don't have the motivation to work as a snake charmer. (Snake Charmer, 30s, male)*

*Due to the snakebite experience, I have started a side job. I work at a garment factory, I'm afraid of snake charming. (Snake Charmer, 20s, male)*

### Discussion

Our research is the first research to clarify the health seeking behavior of the Bede people regarding snakebite and its outcomes considering their lifestyle.

According to two epidemiological studies on snakebite in Bangladesh, the concept of "snake charmer" was not classified independently [12,14]. Furthermore, the actual incidence of snakebite among snake charmers is higher because Bede people were not included in studies

due to existing stigma and discrimination [25]. The research demonstrates that many Bede people were bitten by a snake several times, mainly by the Elapidae. Snake charmer is a common occupation among the Bede due to low levels of education, there are limited options in other careers. As a result, many of them inherit the snake performer profession from their families, where those with less training tend to be bitten by a snake more often than those with more training. A majority of snakebite occur during snake charming performances.

This study revealed that Bede's health seeking behavior for snakebite was influenced by several factors for the utilization of antivenom. Many of the Bede people indicated that they understand the effects of antivenom. However, it is difficult for them to receive antivenom treatment due to four main barriers, accessibility, affordability, availability, and acceptability based on their nomadic style and existing discrimination.

One barrier to using herbal medicine, Indian snakeroot, was also mentioned. This study demonstrated the Bede's strong belief in their herbal medicine, such as Indian snake root (*Rauvolfia serpentina)* based on the mongoose and snake story [26]. This Indian snakeroot may have a detoxification effect on snake venom [27]. However, there is no clear evidence and confirmation about this lore [28]. Other herbs such as flame lily (*Gloriosa superba*) were also used for the treatment. But the Bede's first choice of herb remains to be the Indian snakeroot, backed up by their lore. Despite this popularity, the Indian snakeroot is hard to obtain, as many Bede people mentioned. The Indian snakeroot has been registered with the Convention on International Trade in Endangered Species of Wild Fauna and Flora as on the verge of extinction, As a result, the Bede are beginning to rely on other traditional methods such as tourniquet, incision, and other traditional medicine. All of which have not been proven scientifically to be effective. The Ministry of Health and Family Welfare in Bangladesh has announced that these traditional treatments are ineffective from a scientific viewpoint, or these methods may even be harmful [15]. The application of these traditional methods, along with the low employment of medical treatments, has led to increasing snakebite mortality [29].

The Bede's expertise in snake charming and removing the fangs and venom glands may play a role in reducing the risk of snakebite envenoming, as their snakebite envenoming rate may be lower compared to the general population.

However, there are still instances of snakebite related fatalities among the Bede community each year, as evidenced by six unfortunate deaths in this study.

In this research, only one snake charmer chose to seek modern treatment as the first choice treatment, where he recovered dramatically. In this case, the victim's father-in-law transported him by motorcycle to a medical college hospital in Dhaka. As Sharma et al. have reported, rapid transport of victims decreased the Case Fatality Rate [30]. This is also thought to have saved him. The identification of the snake is also considered to be important in order to perform optimal clinical management treatment and anticipate complications [7]. In this case, the victim was bitten during their snake charm performance. Thus, the snake species identification was possible. It is important to note that a key person is the father-in-law. His decisions and actions saved the victim. Thus, offering such knowledge to the Bede people could be a critical factor in raising the survival chances of snakebite victims.

On the other hand, the six victims resulting in death was unable to seek the appropriate treatment in time. Aside from one unknown case, the victims' condition rapidly worsened, and they died within a few hours after snakebite. These victims received traditional treatment and did not go to the hospital. Neurotoxic symptoms develop quickly, and some victims bitten by cobras were reported to have died within a few hours. Moreover, there has been a case where a victim bitten by a cobra died within eight minutes due to excess catecholamine secretion [31]. Although neurotoxicity is the main symptom, there are various other symptoms even if bitten by a King Cobra [32, 33]. In these cases, victims could not overcome the barriers

of accessibility and availability. If the victims or surrounding people had correct knowledge, the decision for treatment may have been different. An initial visit to a traditional healer and a lack of available transport were all associated with an increased risk of death [28].

## Limitation

This study relied on self reported snakebite incidents, introducing the possibility of recall bias and inaccurate responses. Additionally, recruiting participants through snowball sampling may have led to a sample with similar traits, potentially affecting the generalizability of the findings. Moreover, the limited availability of extensive participant information constrained the thorough analysis of the influence of psychological factors related to the country's complex religious background within the manuscript. Future research should prioritize the collection of more comprehensive participant data to enhance the understanding of these influential factors and their implications on the study's outcomes.

## Conclusion

In conclusion, the Bede community faces significant challenges in accessing timely and appropriate snakebite treatment. Limited successful antivenom cases have been reported, and the community faces barriers in accessibility, affordability, availability, and acceptability of treatment. Traditional treatments are less effective and can lead to fatalities and severe consequences. Collaborations between modern and traditional medicine, training of Bede healers, and inclusion in snakebite programs are recommended to address these challenges.

## Supporting information

**S1 File. Code list.**
(XLSX)

**S2 File. Consolidated criteria for reporting qualitative studies (COREQ): 32-item checklist.**
(DOCX)

**S3 File. Informed consent form.**
(DOCX)

**S4 File. Interview guide for snakebite healer.**
(DOCX)

**S5 File. Interview guide for snakebite victim.**
(DOCX)

**S6 File. Field photos.**
(DOCX)

## Acknowledgments

We would like to thank IACIB's all staff and the Bede people who were involved in this study for their kind support. Specially, we would like to acknowledge the assistance of Mr. Mahammad Ali Khan from IACIB and Mr. Mamun from Bede group, whose support was instrumental in the success of this research.

## Author Contributions

**Conceptualization:** Ken Yoshimura, Moazzem Hossain, Miho Sato, Kazuhiko Moji.

**Data curation:** Ken Yoshimura, Moazzem Hossain, Bumpei Tojo, Paul Tieu, Nathalie Nguyen Trinh, Nguyen Tien Huy, Miho Sato, Kazuhiko Moji.

**Formal analysis:** Ken Yoshimura, Moazzem Hossain, Bumpei Tojo, Miho Sato, Kazuhiko Moji.

**Investigation:** Ken Yoshimura, Moazzem Hossain.

**Methodology:** Ken Yoshimura, Moazzem Hossain, Bumpei Tojo, Miho Sato, Kazuhiko Moji.

**Project administration:** Ken Yoshimura, Moazzem Hossain, Kazuhiko Moji.

**Supervision:** Ken Yoshimura, Moazzem Hossain, Bumpei Tojo, Nguyen Tien Huy, Miho Sato, Kazuhiko Moji.

**Validation:** Paul Tieu, Nathalie Nguyen Trinh, Nguyen Tien Huy, Miho Sato, Kazuhiko Moji.

**Visualization:** Bumpei Tojo, Nguyen Tien Huy, Kazuhiko Moji.

**Writing – original draft:** Ken Yoshimura.

**Writing – review & editing:** Ken Yoshimura, Moazzem Hossain, Bumpei Tojo, Paul Tieu, Nathalie Nguyen Trinh, Nguyen Tien Huy, Miho Sato, Kazuhiko Moji.

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
