## [Decision Letter · Decision Letter 0]

10 Jan 2023

Dear Mr. Yoshimura,

Thank you very much for submitting your manuscript "Barriers to the hospital treatment among Bede snake charmers in Bangladesh: epidemiological survey of snakebite" for consideration at PLOS Neglected Tropical Diseases. As with all papers reviewed by the journal, your manuscript was reviewed by members of the editorial board and by several independent reviewers. In light of the reviews (below this email), we would like to invite the resubmission of a significantly-revised version that takes into account the reviewers' comments. 

We cannot make any decision about publication until we have seen the revised manuscript and your response to the reviewers' comments. Your revised manuscript is also likely to be sent to reviewers for further evaluation.

Sincerely,

Wuelton M. Monteiro, Ph.D.

Section Editor

Wuelton Monteiro

Section Editor

Reviewer's Responses to Questions

**Key Review Criteria Required for Acceptance?**

**Methods**

-Are the objectives of the study clearly articulated with a clear testable hypothesis stated?

-Is the study design appropriate to address the stated objectives?

-Is the population clearly described and appropriate for the hypothesis being tested?

-Is the sample size sufficient to ensure adequate power to address the hypothesis being tested?

-Were correct statistical analysis used to support conclusions?

-Are there concerns about ethical or regulatory requirements being met?

Reviewer #1: The title mentioned an epidemiological survey on snakebites. The methodology is actually qualitative research that uses the snowball method to recruit interviewees within the Bede nomadic community and describe their barriers to seek hospital treatment for snakebite envenoming. I don't see methods relevant for an epidemiological survey and no results that would describe the incidence of snakebites within this particular community.

Reviewer #2: The objective of the study, to “clarify the health seeking behavior of the Bede people regarding snakebite and its outcome”, is clearly articulated in the penultimate sentence of the Introduction. The stated hypothesis, that the study findings should “improve the situations of snakebite among the Bede people as well as…the public health situation in Bangladesh” is testable, but the testing of that hypothesis was not an aim of this study.

As a qualitative work, the study design was appropriate to address the stated objective.

The selection process of individuals to be interviewed was vague, but my understanding is that the researchers employed snowball sampling to recruit a total of 64 interviewees of the Bede people: 38 snakebite charmers who had survived a venomous bite in the previous 3 years, 6 relatives of snakebite charmers who had been bitten in the previous 3 years but had not survived, and 20 snakebite traditional healers. This selection process should be stated more precisely and succinctly. The 64 interviewees were appropriate, given the aim of the study. 

No statistical analysis was performed or required.

On page 9, line 169: neither the paper nor the reference provides sufficient proof that the snakebite victims in this study had experiences “sufficiently traumatic to leave a vivid memory for many years”. This declaration should either be removed or reworded to indicate that the authors presume that the snakebite victims’ experiences were sufficiently traumatic to leave a reliable memory years after the event.

Regarding ethical or regulatory concerns, lines 191-192, “When the participants could not sign their names on the document, we obtained verbal consent from their families”, leave one to wonder why the participant could not give his own consent—unless here we are referring only to deceased victims, in which case this should be stated. Also, what does it mean that “IACIB…offered the Bede free health care”? Free comprehensive health care in perpetuity? During the course of the study? For snakebite treatment only?

Reviewer #3: Study is appropriate for the population of interest. Sample size is small but expected. 

No ethical concerns are present and research team utilized appropriate methods for informed consent and institutional approval.

**Results**

-Does the analysis presented match the analysis plan?

-Are the results clearly and completely presented?

-Are the figures (Tables, Images) of sufficient quality for clarity?

Reviewer #1: In table 2: symptoms: do the authors have any information about symptoms (neurotoxic signs or bleeding) in the group of survivors. 

Under treatment it is mentioned that 30 snakebite victims had treatment at a hospital, but under Snakebite coping and in Figure 2 there are only 5 patients seeking care in a hospital and the majority went for traditional treatment or no treatment. 

In table 1 the authors mention income within the group of 38 survivors. It would be interesting to have numbers about the income within the whole Bede population of 1.2 million and for Bangladesh. Is there a difference?

Reviewer #2: The results are clearly presented on the whole, but there are problems. 

To be consistent and to improve comprehension, the heading on page 11 line 209 should have “survivors” in parentheses after “snake charmers” since the subsequent heading has “deceased cases” in parentheses.

On page 12, line 230, why is this statement inserted here and not under the section labeled Barriers…?

I do not understand the sentence on lines 240-241 of page 12. If I assume that the intention is to say that the individual with full secondary school education was excluded from analysis, I then must ask, “What analysis?” We see the more highly educated person included in Figure 3. We do not see any other analysis.

Table 2. Is there a contradiction in stating that 5 of the 6 death cases occurred during a performance (page 12, line 245), yet in the table, 1 occurred roadside and 4 in the field? Were all these occurrences during performances? 

Table 2. There seems to be a contradiction in Table 2 and Figure 2 in that Table 2 says 30 survivors were treated at hospital whereas Figure 2 shows 5 (4 + 1) were treated at hospital. Can this discrepancy be clarified?

Table 2. Why is the cell that counts symptoms (neurotoxic vs. NA) empty in the column of Survival cases?

Figure 3. The title should insert the word “victims” between “snakebite” and “and”—assuming that this is what this graph is depicting. But does this figure have any real value? Without a control group, we can have no idea whether there is a relevant correlation between education and snake charmers who have survived a snakebite. Would any randomly chosen characteristic of the Bede (odd-number year of birth, for example) give a similar distribution of education level?

The section describing health-seeking behaviors is of great interest, and the use of verbatim quotations is valuable.

Reviewer #3: Results are clearly presented. Table and figures are sufficient. 

I think a map is important. One suggestion would be to add population density and other geographical features of the study site. Another suggestion would be to add a figure of the four venomous species which the snake charmers were bitten by. Added to line 224. Seeing these snakes helps the readers have a visual understanding of the species and possibly the risks of a snake charmer.

**Conclusions**

-Are the conclusions supported by the data presented?

-Are the limitations of analysis clearly described?

-Do the authors discuss how these data can be helpful to advance our understanding of the topic under study?

-Is public health relevance addressed?

Reviewer #1: The authors conclude in the ABSTRACT that the investigation of the health seeking behaviour of the Bede snake charmers may contribute to the improvement of the public health situation in Bangladesh. The Bede are just 1.2 Million of 166 Million people in Bangladesh and they are obviously a special group that earn their income through snake related occupations. It is questionable to infer the entire population from the small Bede community.

The conclusion at the end of the discussion is very confusing: The Bede’s traditional treatments are less effective, although, Indian snakeroot may be effective. From where did the authors get this information and the conclusion that consequently few Bede people die and many suffer from severe consequences? Do the authors have any data on mortality and morbidity in the Bede Community?

Maybe the risk for snakebite envenoming in the Bede Community is even lower compared to the general population, because they are experienced in snake handling and removed the fangs and even the venom glands in many snakes they used during their snake charming performance.

Reviewer #2: Some of the conclusions are supported by the data, but others are not. The final section of the Discussion claims that early, appropriate treatment is essential for the treatment of snakebite. This study was not a quantitative, controlled study. It did not address or test this question, and therefore the researchers have no basis on which to draw this conclusion.

Recall bias and selection bias are listed as the study’s two limitations. This is valid.

The authors discuss how their data should be helpful to the advancement of our understanding of snakebite treatment behavior among the Bede. By drawing attention to the importance of traditional practices and beliefs in this society, the paper’s public health relevance extends, in spirit if not in detail, to other strongly traditional settings where snakebite is common.

Reviewer #3: Conclusions are supported by the collected data. Limitations addressed and some sampling bias exists. 

Authors discuss the importance of the findings for the population. I would suggest adding some possible interventions that could be implemented for improved awareness campaigns for Bede snake charmers and others living in this region of Bangladesh.

**Editorial and Data Presentation Modifications?**

Reviewer #1: (No Response)

Reviewer #2: A great deal of copy editing to correct grammatical and other linguistic errors is needed. 

The article is broader in scope than the title suggests. Only a small part of the paper deals with barriers to hospital treatment of snakebite among Bede snake charmers. The paper also covers barriers to traditional treatment, sociodemographic characteristics of snake charmers, preventive measures practiced by snake charmers, and methods of snakebite treatment employed by traditional Bede healers. The paper’s title should reflect the content of the work.

On page 6, lines 92-93 and 95-96 are repetitious. Similarly lines 102-104 and 105-106 are repetitious.

On page 12, lines 245 and 250: since there were only 6 death cases, if 5 died during a performance, the 1 who did not should not be referred to as “the remainders”. If 5 victims died within a few hours after the snakebite, only 1 did not. That 1 should not be referred to as “the remaining cases” or “the victims”.

There is considerable inconsistency between headings and subsequent text. For example, a heading on page 16, line 274, is “Barriers influencing the health seeking behavior” yet it is not until page 20, line 420 that the paper begins to cite barriers to health seeking behavior. Lines 278-304 present risk factors; lines 306-344 present preventive measures; lines 346-382 present diagnosis; lines 385-412 describe symptoms and treatment.

Lines 414-418 belong under Availability.

Line 450 also probably belongs under Availability.

Since the preceding pages discussed barriers to hospital treatment, a new heading before line 524 should be inserted to indicate that the ensuing text discusses barriers to traditional treatment.

Page 21, line 422. “Accessibility” in public health is generally used to connote a combination of factors including geography, affordability, and availability. Although it puts a small dent in the “4 A’s”, “accessibility” in this paper is being used more in the sense of geographic accessibility and would be better labeled as such.

Reviewer #3: (No Response)

**Summary and General Comments**

Reviewer #1: The authors present the results of interviews with 38 snakebite victims, who survived the snakebite mostly during snake handling and 6 snakebite victims, who died after a bite. Although an epidemiological survey is mentioned in the title and the first part of the discussion, there are no results of such a survey. The barriers Availability, Accessibility, Affordability and Acceptability are common barriers for most snakebite victims in many regions in Asia and Africa and not really new or special for this certain community of snake charmers and handlers.

Reviewer #2: This study is informative, original, and a welcome addition to our knowledge of snakebite. As written, readers’ comprehension suffers due to inadequate organization and mangled English. I hope the paper can be reworked to make it worthy of publication in PLOS.

Reviewer #3: Manuscript was well written, and investigation is sound for the population of interest. Only minor revisions or clarifications needed. One suggestion that I think is important is adding "venomous snakebite" to the title and "venomous snakebite" to other section headings on line 219, 242 and 268. I would ensure throughout the manuscript the "venomous snakebite" is added, like in table 2 and figure 2. Despite not knowing fully if an individual was "envenomated" we do know they were venomous snakes, and as such I believe this is important to highlight in these key areas of the manuscript. Six snake charmers were excluded because they sustained nonvenomous snakebites and the emphasis of this investigation is Bede snake charmers who handle and work with venomous snakes. Overall, excellent manuscript.

PLOS authors have the option to publish the peer review history of their article (what does this mean?). If published, this will include your full peer review and any attached files.

Reviewer #1: No

Reviewer #2: Yes: Ellen M. Einterz, MD, MPH&TM

Reviewer #3: Yes: Norman L. Beatty, University of Florida College of Medicine, Gainesville, FL, USA
---

## [Decision Letter · Decision Letter 1]

26 Apr 2023

Dear Yoshimura,

Thank you very much for submitting your manuscript "Barriers to the hospital treatment among Bede snake charmers in Bangladesh with special reference to venomous snakebite" for consideration at PLOS Neglected Tropical Diseases. As with all papers reviewed by the journal, your manuscript was reviewed by members of the editorial board and by several independent reviewers. In light of the reviews (below this email), we would like to invite the resubmission of a significantly-revised version that takes into account the reviewers' comments. 

We cannot make any decision about publication until we have seen the revised manuscript and your response to the reviewers' comments. Your revised manuscript is also likely to be sent to reviewers for further evaluation.

Sincerely,

Wuelton M. Monteiro, Ph.D.

Section Editor

Wuelton Monteiro

Section Editor

Reviewer's Responses to Questions

**Key Review Criteria Required for Acceptance?**

**Methods**

-Are the objectives of the study clearly articulated with a clear testable hypothesis stated?

-Is the study design appropriate to address the stated objectives?

-Is the population clearly described and appropriate for the hypothesis being tested?

-Is the sample size sufficient to ensure adequate power to address the hypothesis being tested?

-Were correct statistical analysis used to support conclusions?

-Are there concerns about ethical or regulatory requirements being met?

Reviewer #2: Lines 168-173: 

• Do you mean: …38 adult Bede (survival cases) as well as 1 relative of each of the 6 deceased Bede snake charmers who died due to venomous snakebite…? If so, can this be stated with that clarity?

• "6 nonvenomous snakebite victims were excluded…" Excluded from what number? If not from the 38, can you give give us the total number of snakebite victims, the pool from which you selected your interviewees? Did you start with 44 snakebite survivors? Or with 50 snakebite victims?

Reviewer #3: Objectives of study are clarified with revised submission. Description of population is clear. 

No ethical concerns are present.

Reviewer #4: This article is not clear and clear, and seems to have great risks in ethics and morality. In short, there are major problems and they should be amended or rejected.

**Results**

-Does the analysis presented match the analysis plan?

-Are the results clearly and completely presented?

-Are the figures (Tables, Images) of sufficient quality for clarity?

Reviewer #2: I did not find that Figure 2, added as a clarification to explain participants in the study, was helpful. In fact, it is not readily understandable, and I would recommend it be removed. 

In Table 2, “Best treatment”, which you intended to be more explanatory, is meaningless as it stands. If I am understanding your answer to my comments to the original version, “Best treatment” should be replaced by “What do you think is the best treatment for snakebite?” or something to that effect.

Lines 280-281: This belongs in Methods, not Results.

There are too many sub-headings in the Results section, and since the text that follows each sub-heading does not always adhere to what the sub-heading is signaling, I would suggest removing most of those sub-headings. Could you pare them down to three? These could be “Venomous snakebite experience of snake charmers (survivors)”; “Venomous snakebit experience of snake charmers (death cases)”; “Key informant interview comments”. In the key informant interview comments section, you could simply summarize with a line of text the one or multiple quotations that follow for each idea.

Reviewer #3: Results are clear. Authors have addressed each comment or request from the Reviewer panel.

Reviewer #4: It's not clear and complete. I even suspect that the author did not write this part at all, the author just laid out some information, meaningless.

**Conclusions**

-Are the conclusions supported by the data presented?

-Are the limitations of analysis clearly described?

-Do the authors discuss how these data can be helpful to advance our understanding of the topic under study?

-Is public health relevance addressed?

Reviewer #2: Likewise, I would suggest not subdividing the Discussion with sub-headings, since again the text after each sub-heading does not adhere closely enough to what is being signaled by the sub-headings.

Reviewer #3: Data is presented in a concise manner.

Reviewer #4: Too bad the author's description is not clear enough to support the conclusion. Even our argument that snakebite is not a tropical disease does not fit the purpose of the journal.

**Editorial and Data Presentation Modifications?**

Reviewer #2: Line 51: “Essential” is untested and unproven; I would suggest that another word be used.

Lines 621-635 beginning with “One barrier to the utilization of herbal medicine…” and ending with “…has led to increasing snakebite mortality [29].” should be a separate paragraph.

Reviewer #3: (No Response)

Reviewer #4: N/A

**Summary and General Comments**

Reviewer #2: I am grateful to the authors and editors for this revision. Most of my comments were addressed, and some changes were made that improve the paper. 

I congratulate the authors on a well-researched, thoughtful study and encourage them to make every effort to improve it. Careful presentation to ensure optimal comprehension is important, not least because the topic of this paper is important.

Reviewer #3: Overall, the manuscript is well written and revised manuscript is presenting interesting findings. Authors have addressed each Reviewer's comment or recommendation. In my opinion the manuscript is ready for publication with some minor editorial or grammatical corrections.

Reviewer #4: There is no doubt that this manuscript pays attention to social hot issues and uses more correct research methods. However, the description of the methods is relatively thin, which is difficult for our readers to understand and reproduce, which restricts our analysis of the value of this manuscript.

1.I would like to question whether the team of researchers can actually support such a study, whether other people are involved in the work but do not declare it, and whether they need to add the specific contribution of the author, preferably in detail; or why Japanese research institutions, with only a few researchers from Bangladesh, have access to primary data on Bangladesh. In which communication on Earth was the author team working at the time of the project?

2.The data came from Bangladesh. Why is it a global problem? Do people in other countries have regular contact with snakes? The expandable aspects of this study seem to have been somewhat enlarged, and it is hoped that the authors will be able to make the necessary modifications. This seems to be a "regional issue".

3.The author needs to adjust the logic of the article and study whether the treatment is traditional, ethnic, or modern, biotechnology-based. Sentences are a bit of a mishmash.

4.It seems that some of the words used are too trendy for easy reading.

5.In general, all data, as long as it is not obtained from one's own experiment, needs to be supported by literature sources. The source of much of the data in this manuscript is questionable, and the authors should correct it against references.

6.As an article published in a journal, the author of this article mentions a great deal of material unrelated to the main idea. Authors should enhance the relevance of the main idea or remove part of the content to avoid misleading readers and provide effective information. A lot of basic information is especially unnecessary.

7.Does the work reported in the manuscript have relevant ethical approval? I seem to see several references to inappropriate ethical requests throughout the text. How many different institutions do the authors come from? What are the ethical requirements of other institutions? Is there a problem here? It needs to be explained.

The author did not control variables well. In a country with a complex religious background, many factors will affect the selection of patients and even the rehabilitation of patients in terms of psychological factors. The author appears to briefly mention these issues but does not provide any further elaboration based on them. I hope to pay attention to similar issues.

8.There is a problem with the structure of the article. A lot of the content written in "results" should be "methods". Authors are encouraged to submit the manuscript structure to their peers in other work groups prior to submission.

9.The authors should personally attest to the appropriateness of the literature.

10.The report of the results is too flat and direct, lacking logic, and easy to understand. 11.How does this information relate to the main idea? No discourse is seen, even in the following text.

12.part of the spelling problem.

13.The snake-catching scene doesn't make sense. The author's survey is based on the self-description of the population concerned, and the author also mentions that they do not have a high level of education. Can they exclude the implausible or exaggerated parts of the description? No embodiment of these methods is seen. There seems to be a need for clarification.

14.Were all the included subjects from Bangladesh? Are there exclusion criteria? It needs to be confirmed.

15.The math and statistics seem to be wrong.

16.Can you provide pictures from the research phase to prove the source of the data? 17.Is there an intuitive picture that describes the content of this article?

18.The necessary cultural background is not stated, and as an international journal, this can affect readers' understanding. There are too many topics that people in other countries and regions do not understand, so they should be more specific.

19.Are the people included in the study a single factor? Or is it influenced by more factors? This is related to the reliability of this paper. The authors did not report specific analyses, and the conclusions are far-fetched.

20.How were the conclusions reached when the authors did not analyze the data at all? 21.How do I prove it? The problem is too serious.

22.Concepts are not clearly defined in the last few sections of the manuscript.

23.Conclusions are arbitrary, lack evidence, and need to be supplemented.

24.The contrast is confusing, and I don't know what I want to say.

25.The author completely failed to face up to and modify the comments and suggestions of the previous reviewers and only made some modifications unrelated to the center, which is meaningless. There seems to be a problem with the author's academic attitude.

26.The contents of nationality disputes should be properly handled.

In conclusion, this manuscript is too crude and needs to be revised. It should not even be accepted for publication.

PLOS authors have the option to publish the peer review history of their article (what does this mean?). If published, this will include your full peer review and any attached files.

Reviewer #2: Yes: Ellen M. Einterz, MD

Reviewer #3: Yes: Norman L. Beatty, MD, University of Florida College of Medicine

Reviewer #4: Yes: Jing Zhang
---

## [Editor Report · Decision Letter 2]

7 Aug 2023

Dear Yoshimura,

We are pleased to inform you that your manuscript 'Barriers to the hospital treatment among Bede snake charmers in Bangladesh with special reference to venomous snakebite' has been provisionally accepted for publication in PLOS Neglected Tropical Diseases.

Best regards,

Wuelton M. Monteiro, Ph.D.

Section Editor

Wuelton Monteiro

Section Editor

---

## [Editor Report · Acceptance letter]

14 Sep 2023

Dear Yoshimura,

We are delighted to inform you that your manuscript, "Barriers to the hospital treatment among Bede snake charmers in Bangladesh with special reference to venomous snakebite," has been formally accepted for publication in PLOS Neglected Tropical Diseases.

Best regards,

Shaden Kamhawi

co-Editor-in-Chief

Paul Brindley

co-Editor-in-Chief
